# Saponins as Modulators of the Blood Coagulation System and Perspectives Regarding Their Use in the Prevention of Venous Thromboembolic Incidents

**DOI:** 10.3390/molecules25215171

**Published:** 2020-11-06

**Authors:** Beata Olas, Karina Urbańska, Magdalena Bryś

**Affiliations:** 1Department of General Biochemistry, Faculty of Biology and Environmental Protection, University of Lodz, Pomorska 141/3, 90-236 Lodz, Poland; 2Faculty of Medicine, Medical University of Lodz, 90-419 Lodz, Poland; karina.urbanska@stud.umed.lodz.pl; 3Department of Cytobiochemistry, Faculty of Biology and Environmental Protection, University of Lodz, Pomorska 141/3, 90-236 Lodz, Poland; magdalena.brys@biol.uni.lodz.pl

**Keywords:** saponins, thromboembolic incidents, antiplatelet activity, antithrombotic effect, anticoagulation

## Abstract

Saponins comprise a heterogenous group of chemical compounds containing a triterpene or steroid aglycone group and at least one sugar chain. They exist as secondary metabolites, occurring frequently in dicotyledonous plants and lower marine animals. Plant saponin extracts or single saponins have indicated antiplatelet and anticoagulant activity. Venous thromboembolism (VTE), including deep venous thrombosis and pulmonary embolism, is a multifactorial disease influenced by various patient characteristics such as age, immobility, previous thromboembolism and inherited thrombophilia. This mini-review (1) evaluates the current literature on saponins as modulators of the coagulation system, (2) discusses the impact of chemical structure on the modulation of the coagulation system, which may further provide a basis for drug or supplement design, (3) examines perspectives of their use in the prevention of VTE. It also describes the molecular mechanisms of action of the saponins involved in the prevention of VTE.

## 1. Introduction

A number of plants produce saponins as secondary metabolites. They are particularly common in families such as the *Araliaceae*, *Leguminosae*, *Polygalaceae*, *Campanulaceae*, *Dioscoreaceae*, *Liliaceae*, and *Scrophulariaceae* [1,2]. Many of these plants have long been used in alternative medicine, such as *Panax notoginseng* (Burkill) F.H.Chen, *Bupleurum chinense* DC., *Paris polyphylla* Sm., and *Dioscorea polystachya* Turcz. in Chinese medicine. However, specialized research equipment is needed to accurately determine the composition of these plants and study the pharmacological action of individual chemical compounds [3,4].

It is difficult to incorporate the saponins present in terrestrial plants, and some marine organisms, in modern pharmaceuticals. The saponin content in plants depends on many factors, including the plant species and organ, as well as environmental factors and post-harvest treatments. Individual plant species usually contain a mixture of many saponins with a wide variety of chemical and biological properties [5,6].

Saponins are amphipathic glycosides that exhibit foaming properties. Structurally, they consist of a lipophilic polycyclic aglycone, termed a sapogenin, attached to one or more sugar side chains. Figure 1 contains a diagram showing the structure of saponins and their potential properties for preventing thromboembolic events [7,8,9].

Saponins have been attributed to various health benefits, including cholesterol reduction, antioxidant activity, and cancer risk reduction. For example, cardioprotective properties, including anti-atherosclerotic properties of 20(S)-protopanaxadiol saponins, have been observed in ApoE-deficient mice [10]. It has been found that the administration of saponins isolated from leaves of *Panax quinquefolius* L. (125 and 250 mg/kg daily, for 15 days) influences cisplatin-induced cardiotoxicity, in part by inhibition of nuclear factor kappa-light-chain-enhancer of activated B cells (NF-κB) activity and regulation of the phosphoinositide 3-kinase (PI3K)/protein kinase B (Akt)/apoptosis mediated signaling pathway. The tested saponins also reduced the level of cisplatin-induced oxidative stress in mice, as determined by reactive oxygen species concentration, glutathione level and superoxide dismutase activity [11]. Similar properties have been observed for a steroidal saponin extract from *Ophiopogon japonicus* (Thunb.) Ker Gawl. root: This extract was shown to alleviate chronic doxorubicin-induced heart failure in an animal model by inhibiting oxidative stress [12]. Sun et al. and Dong et al. also report *Panax notoginseng* saponins to demonstrate antioxidative activity [13,14]. However, the chemical structure, and hence the properties of saponins, have been found to change during storage, and under other conditions [7].

Venous thromboembolism (VTE), including deep venous thrombosis and pulmonary embolism, is a multifactorial disease whose occurrence is strongly influenced by characteristics such as patient age, immobility, previous thromboembolism and inherited thrombophilia. Other risk factors for VTE include obesity, hospitalization for surgery, active cancer, confinement in a nursing home, immobility or leg paresis, oral contraception and hormone therapy, among others. Moreover, VTE is categorized by the U.S. Surgeon General as a major public health problem [15,16].

This mini-review evaluates the current literature on saponins as modulators of the coagulation system, and perspectives regarding their use in the prevention of VTE. It also describes the mechanisms of action of saponins involved in the prevention of VTE.

## 2. Saponins with Antiplatelet and Anticoagulation Activity

Studies on the presence of antiplatelet and anticoagulant compounds in plants began in the 1980s. It has since been found that a variety of groups of chemical compounds are responsible for this anti-platelet activity, with terpenoids and flavonoids being the main class of active compounds in over 30% of studied plants, followed by saponins, phenylpropanoids, coumarins and various others [17]. Later published research on antiplatelet and anticoagulant activity has been performed with both the use of plant extracts and/or pure compounds [3,11,18,19].

Some studies report the pharmacological properties of single and specific saponins with antithrombotic, anti-platelet aggregation and anticoagulation effects [20,21,22,23]. Table 1 summarizes the results of recent in vitro and in vivo studies, based on various models, on the antiplatelet and antithrombotic activity of individual saponins. For example, some saponins have an effect on the coagulation cascade, including prothrombin and other coagulation factors [18,23].

## 3. The Molecular Mechanism of Saponin Action as Modulators of the Coagulation System

### 3.1. Arachidonic Acid Pathway

Arachidonic acid (AA) is released from the phospholipid molecules of the cell membranes by the action of phospholipase A_2_ (PLA_2_); this is important for the formation of prostanoids, which play a role in platelet activation and inactivation mechanisms [43]. Arachidonic acid (AA) is the preferred substrate for the constitutive cyclooxygenase enzyme (COX-1) that enables the production of prostaglandin endoperoxides (PG). COX-1 converts AA into two transient endoperoxides, prostaglandin G_2_ (PGG_2_) and prostaglandin H_2_ (PGH_2_). These compounds are then converted by thromboxane synthetase to thromboxane A_2_ (TXA_2_), which is the major metabolite in platelets [44]. TXA_2_ is rapidly and nonenzymatically converted to inactive and stable thromboxane B_2_ (TXB_2_). Thromboxane A induces shape changes and aggregation among platelets, as well as their degranulation [45].

TXA_2_ affects cellular signal transmission by interacting with a specific protein receptor located on the cell surface. This thromboxane receptor (TP) is encoded by the *TBXA2R* gene and belongs to the G protein-coupled superfamily of seven-transmembrane receptors [46]. As TXA_2_ is known to participate in the pathogenesis of arterial thrombosis, a number of anti-TXA_2_ agents have been developed to inhibit its biosynthesis or act as TP antagonists. The process of platelet activation is regulated by various factors, one of which is the level of the secondary messenger’s adenosine 3’, 5’-cyclic monophosphate (cAMP) and guanosine 3’,5’-cyclic monophosphate (cGMP): Increases in the intracellular levels of cAMP and cGMP result in inhibition of agonist-induced platelet activation, aggregation and adhesion. At the same time, arachidonate 12-lipoxygenase (12-LOX) converts AA into 12-hydroxy-5,8,10,14-eicosatetraenoic acid (12-HETE) [43,44,47].

The ginsenoside Rk1 **(24)** isolated from *Panax ginseng* C.A.Mey. has been found to demonstrate a stronger response to AA-induced platelet aggregation than acetylsalicylic acid. Rk1 is known to reduce the levels of TXB_2_ and 12-HETE. It is believed that Rk1 modulates intracellular calcium level, known to induce platelet activation, by increasing 12-LOX translocation from the cytosol to the membrane; this movement reduces the level of Ca^2+^, thus decreasing that of 12-HETE [48].

The steroidal saponin timosaponin AIII (TAIII) [3-O-β-d-glucopyranosyl-(1→2)-β-d-galactopyranoside sarsasapogenin] **(6)** has also been found to inhibit the TXA_2_ receptor. TAIII, first isolated from *Anemarrhena asphodeloides* Bunge, has been found to inhibit platelet aggregation by suppressing ADP secretion, regardless of the increase in cAMP and cGMP; it has also been found to inhibit the production of TXA_2_ in platelets. TAIII has also been shown to inhibit Gq rather than G_12/13_ activation through the thromboxane receptor [21].

### 3.2. GPVI Signaling Pathway and Enzymatic Cascades

The binding of collagen to the glycoprotein VI (GPVI) receptor causes the phosphorylation of the γ-chain tyrosine of the Src family kinase-mediated Fc receptor, followed by the tyrosine of spleen tyrosine kinase (Syk) and that of linker for activation of T cells (LAT). On the cell surface, GPVI may be expressed as monomeric and dimeric forms, with a single GPVI molecule associating with a single Fc receptor γ-chain dimer [49]. Subsequent activation of LAT and PI3K mediates the recruitment and tyrosine phosphorylation of phospholipase Cγ2 (PLCγ2); this results in 1,2-diacylglycerol (DAG) and inositol-1,4,5-trisphosphate (IP3) release, protein kinase C (PKC) activation, and Ca^2+^ mobilization.

Such platelet activation affects the release of fibrinogen and P-selectin, together with various prothrombotic factors, such as thrombin, adenosine diphosphate (ADP) and TXA_2_. Mitogen-activated protein kinases (MAPKs), such as ERK-2, p38-MAPK and JNK-1, are also activated by agonist-stimulated platelets. It is suggested that collagen-induced activation of PI3K and PLC may be upstream events mediating ERK-2 and p38-MAPK phosphorylation [36,49,50,51].

It was found that dihydroginsenoside Rg3 **(11)** from *Panax ginseng* may suppress the phosphorylation of ERK-2, but not that of p38-MAPK during collagen-stimulated aggregation. As the intracellular concentration of Ca^2+^ is often found to increase as a result of such stimulation, it is possible that its level may play a role in collagen stimulation [31]. Panaxatriol saponins (PTS), isolated from an extract of *Panax notoginseng* known to demonstrate antiplatelet activity, have also been found to suppress cellular signaling by reducing ERK-2 and p38 phosphorylation [22].

Ginsenoside-Rp1 (3-O-β-d-glucopyranosyl-(1→2)-β-d-glucopyranosyl dammarane-3β, 12β-diol) **(26)** inhibits collagen-induced platelet activation and thrombus formation through modulation of early GPVI signaling events; this effect involves the stimulation of vasodilator-stimulated phosphoprotein (VASP), and ERK-2 and p38-MAPK [36]. VASP is a major substrate for cAMP-dependent protein kinase A (PKA) and cGMP-dependent protein kinase G (PKG) and is an actin- and profilin-binding protein expressed in platelets [52].

Another saponin that affects platelet aggregation by exerting an influence on cyclic cAMP and cGMP nucleotides is dihydroginsenoside Rg3, a relatively stable chemical derivative of the triterpene saponin ginsenoside Rg3 isolated from *Panax ginseng*. Interestingly, dihydroginsenoside Rg3 has been found to elevate cyclic AMP production in resting platelets, but not cyclic GMP production. It is possible that matrix metalloproteinase and vasodilator-stimulated phosphoprotein play a role in this process [31].

Ginsenoside-Rp1 is known to increase the cAMP level and suppress collagen-induced ATP-release, thromboxane secretion, p-selectin expression, Ca^2+^ mobilization and integrin α_IIb_β_3_ activation; it has also been found to reduce p38-MAPK and ERK-2 activation and inhibit tyrosine phosphorylation of many kinases (Fyn, Lyn, Syk, LAT, PI3K and PLCγ2) of the GPVI signaling pathway. Ginsenoside-Rp1 was also observed to inhibit in vivo thrombus formation and ex vivo platelet aggregation and ATP secretion without interfering with coagulation time [36]. In turn, ginsenoside Rp3 **(27)** modulates agonist-induced platelet activation and thrombus formation by integrin α_IIb_β_3_ activation, MAPK signaling, Src, PLCγ2 and PI3K/Akt activation, and VASP stimulation [37]. Additionally, ginsenoside Ro **(25)**, an oleanane-type saponin, inhibited thrombin-induced platelet aggregation and reduced the binding of fibrinogen to αIIb/β3 via cAMP-dependent vasodilator-stimulated phosphoprotein phosphorylation. In addition, ginsenoside-Ro abrogated clot retraction reflecting the intensification of thrombus [53].

Notoginsenoside Fc **(37)** is a novel saponin isolated from *Panax notoginseng*. The phospholipase C-γ 2 has been proposed as the focal point of notoginsenoside Fc antiplatelet activity, which is believed to act through the downregulation of DAG, protein kinase C, TXA2 and IP3. In contrast, the notoginsenoside Ft1 is believed to promote platelet aggregation by potentiating signaling by the PLCγ2-IP3/DAG-(Ca^2+^)/PKC-TXA2 pathway **(38)**. Ft1 is isolated from *Panax notoginseng* in the same way as notoginsenoside Fc [40].

The main components of the arachidonic acid and GPVI signaling pathways, discussed above, are shown in Figure 2. Several of the saponins listed in Table 1 have the ability to inhibit ADP-induced platelet aggregation. These compounds are therefore antagonists of ADP-activated transmembrane receptors such as P2Y_1_ and P2Y_12_. Both of these receptors interact with G proteins, but their downstream effectors are in different signaling pathways. The activation of a P2Y_1_ ADP receptor, which belongs to the Gq protein-coupled receptors, leads to the activation of phospholipase C (PLC), whereas activation of the P2Y_12_ receptor coupled to the Gi protein triggers adenylate cyclase activity. It has been suggested that activation of both receptors is required for a complete platelet response to ADP [54,55]. Figure 2 shows the potential effectors inhibited by ADP receptor inhibitors.

### 3.3. Inhibition of Tissue Factor Expression

Tissue factor (TF) is the key element in triggering the primary procoagulant pathway of coagulation and is an important factor associated with the state of hypercoagulability in venous thrombosis. TF dysfunction may result in the occurrence of pulmonary embolism, chronic venous insufficiency or post-thrombotic syndrome, among others [56]. TF is a transmembrane protein acting as a high-affinity nonenzymatic cofactor for factor (F)VIIa in the extrinsic tenase (TF-FVIIa) complex, acting in turn as an activator of the coagulation protease cascade. TF stabilizes the FVIIa catalytic site on the plasma membrane, allowing optimal interaction with substrates FIX and FX [57,58,59].

TF is produced by macrophages and microvesicles in atherosclerotic plaques and activated cells within the vasculature, as well as perivascular cells, all of which are known to activate coagulation. It is expressed in an organ-specific manner, being observed at high concentrations in the brain, lung, heart, uterus and placenta, and low concentrations in skeletal muscles and joints. This phenomenon suggests that TF provides additional hemostatic protection to a selected group of important organs. However, TF-dependent activation of blood coagulation needs to be tightly regulated to maintain hemostasis and prevent thrombosis; the first level of this regulation is believed to be based on the management of TF gene/protein expression [56].

It has been shown that the natural D39 **(3)** steroidal saponin isolated from *Liriope muscari* (Decne.) L.H. Bailey downregulates endothelial TF expression and venous thrombus formation by modulating the PI3K/Akt/GSK3β and NF-κB signaling pathways. It has been proposed that D39 exerts this activity by binding to, and thus deactivating, NMMHC IIA (non-muscular myosin heavy chain IIA), and inhibiting the dissociation of NMMHC II2 from tumor necrosis factor receptor 2 (TNFR2) [60,61].

## 4. Toxicology and Safety of Saponins

The rate of absorption of saponins in the human digestive tract is very low, except for those whose glycosyl groups are hydrolyzed. This process can occur under natural conditions through heating (physically), acidolysis (chemically) and through microbiological transformation. The intestinal flora exerts a strong metabolic effect on saponins, mainly through the hydrolysis of sugar [7]. For example, the oral bioavailability of ginsenosides Ra3 **(16)**, Rb1 **(17)** and Rd **(19)** was found to be 0.1–0.2%, and 0.2–0.6% for the ginsenosides Re **(23)**, Rg1 **(20)**, and notoginsenoside R1 **(39)** [62].

Only six saponin drugs have been registered to the DrugBank database [63] (version 5.1.4, accessed in January 2020). The first is aglycone Smilagenin [(25R)-5β-spirostan-3β-ol)], and the next five are various other ginsenosides. In the case of ginsenoside C, it has been noted that the risk of thrombosis may be associated when administered in combination with other drugs.

A pharmacological interaction has also been demonstrated between saponins from *Panax notoginseng* and aspirin regarding the absorption process; however, the research was conducted on an animal model. A showed a marked increase in the absorption of notoginsenoside R_1_, ginsenosides Rg1, Rb1, Re and Rd was observed when aspirin was present [64,65].

Few studies have examined the toxic doses associated with saponin use. Nevertheless, these compounds have been found to be are highly toxic when administered intravenously in higher animals; however, this toxicity falls to low levels when administered orally. Some saponins present in food do not possess significant oral toxicity [66]. A study of the sub-acute and chronic toxic effects of total steroidal extracts from *Dioscorea zingiberensis* C.H. Wright on internal organs and biochemical indicators found that dogs could tolerate the extracts at doses up to 500 mg/kg with LD_50_ greater than 3000 mg/kg body weight [67]. Due to their toxic effects on some mammals, saponins are proposed as rodenticides [68,69]. The oral LD_50_ of total saponins and tannins from *Dialium guineense* Willd. stem bark has been found to be greater than 5000 mg/kg body weight for male Sprague–Dawley rats [70], while that observed for saponins extracted from the rind of *Citrullus colocynthis* (L.) Schrad. was found to be 79.43 mg/rat for Narway rats [71]. Saponins isolated from *Albizia julibrissin* Durazz have also demonstrated reproductive toxicity for female mice, most likely caused by damage to the ovary and uterus [72].

Some triterpene and steroid saponins are unsuitable for medical applications due to their harmful levels of hemolytic toxicity. Such activity is known to depend on both the type of aglycone group and the number and sequence of sugar side chains. Presumably, saponins interact with the cholesterol in the erythrocyte membrane, which results in pore formation, the disruption of the ionic balance and membrane permeabilization [73,74]. de Groot et al. report that α-tomatine is a promising saponin in that it demonstrates quite low hemolytic activity; however, it has a high affinity for cholesterol [75]. Sarykayha et al. report that saponins with oleanolic acid in the aglycone group cause significant hemolysis compared to those with a hederagenin aglycone: Aristatoside C, cephoside A and davisianoside B exhibited significant hemolytic activity on human blood cells when administered at 500 μg/mL, the respective values being 89.80%, 67.70% and 66.17% [76].

A study of 41 triterpenoid saponins and sapogenins found that many structure-dependent factors modulate the hemolytic activity, such as skeleton type, location of functional groups and the stereochemical configuration of substituents in aglycone, as well as the complexity of sugar moieties. The presence of polar regions on sapogenins, such as a carboxyl group at position 28, an α-hydroxyl group at position 16, and/or a β-hydroxyl group at position 2, significantly enhanced hemolysis. In contrast, the presence of an α-hydroxyl group at position 2 or a hydroxymethyl group at positions 23 or 24 was associated with the decreased hemolytic activity. Additionally, it turned out that the majority of oleanane-type sapogenins showed stronger hemolytic effects than those of the ursane and dammarane types [77].

A modern and safe solution to the problem of saponin toxicity may lie in the organic synthesis of diverse saponins with therapeutic effects accompanied by the systematic study of their biological properties and toxicity [5,78]. Even more effective are in silico studies that allow therapeutic saponins to be designed without prior organic synthesis. One such study was performed by Zheng et al. Briefly, two groups of saponins were selected based on literature data, one with hemolytic properties and the other without, and a machine-learning-based hemolytic toxicity classification model was created for them. The study resulted in the design of an innovative program called “e-Hemolytic-Saponin” for the automatic prediction and design of hemolytic/non-hemolytic saponins. The program allowed the analysis of 452 saponins and the creation of a database of hemolytic activity as a guide for the design and synthesis of more non-hemolytic saponins with therapeutic effects [73].

## 5. Conclusions

In recent years, the structure, distribution, and pharmacological importance of saponins have been extensively analyzed, both as a pure compound and as an enriched crude plant extract. Literature data clearly indicate that these compounds have a high anticoagulant/antithrombotic potential that can be used in the prevention and treatment of thromboembolic disease; both in vitro and in vivo models have found steroidal and triterpenoid saponins isolated from various plants to have these properties [18,79,80,81]. Xiong et al. report that *P. notoginseng* root and rhizome (0.5–3.5 mg/mL) demonstrate anticoagulant activity in vitro [11]. Zhang et al. also note that saponins isolated from the roots and rhizomes of *P. notoginseng* may play an important function in treating cardiovascular diseases such as coronary heart diseases, cardiac arrhythmia and angina pectoris [79]. In addition, Dong et al. found *P. notoginseng* saponins to protect human umbilical vein endothelial cells from H_2_O_2_-stimulated oxidative stress [14].

The broad-spectrum anticoagulant/antithrombotic effects of saponins could be derived through several mechanisms of action. For example, the anti-platelet mechanisms of ginsenosides such as Rg3, Rp1, and Rp3 have been associated with elevated cAMP, integrin α_IIb_β_3_ activation, granule secretion, and Ca^2+^ mobilization; however, other mechanisms may exist for saponins with different structures.

Platelets are activated in many ways, so even with efficient inhibition of platelets by ADP inhibitors such as clopidogrel, prasugrel, or ticagrelor, their activation may occur through other molecular pathways [81]. Whereas, saponins as a group of chemical compounds offer broad anticoagulant activity through various cell signaling pathways.

However, saponins also demonstrate unfavorable side effects when consumed, and so careful strategies are needed to create compounds for clinical use. Problems associated with the low oral bioavailability of saponins need to be eliminated. Another important issue is to reduce or completely eliminate the hemolytic activity of saponins intended for use as anticoagulants.

The ability of saponins to increase membrane permeabilization is also an important issue. On the one hand, it can increase the bioavailability of drugs or vitamins, and on the other hand, it may lead to pro-thrombotic activity. This is especially important in the prothrombin activation process in which vitamin K serves as an essential effector for γ-glutamyl carboxylase, an enzyme that catalyzes the carboxylation of glutamic acid residues in prothrombin. Saponins are substances that increase the permeabilization of the microsomal membrane in this process and thus facilitate the coagulation [82,83,84].

Due to the lack of knowledge of their properties and their complex chemical composition, plants do not appear completely suitable for use in modern medicine. For example, different notoginsenosides isolated from the same plant of the genus *Panax* demonstrate quite different anticoagulant effects. Therefore, the best course of action may lie in the organic synthesis of new saponins, preceded by in silico analyses and modeling. Based on the already available knowledge about the influence of the chemical structure of saponins on their biological properties, one can strive to create appropriate compounds which are both non-toxic yet pharmacologically effective.

## Figures and Tables

**Figure 1 molecules-25-05171-f001:**
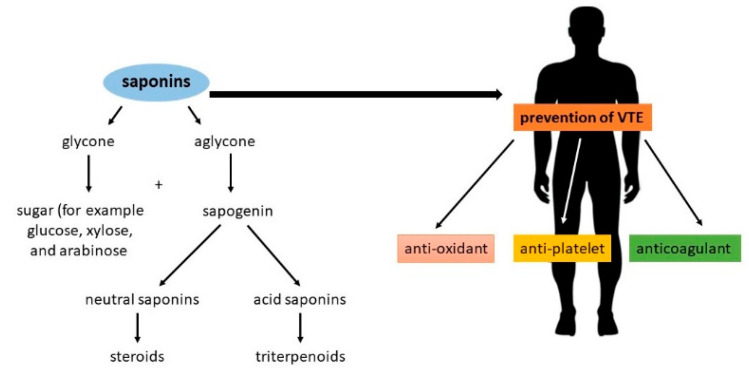
The structure of saponins and their potential properties in the prevention of venous thromboembolism [8,9].

**Figure 2 molecules-25-05171-f002:**
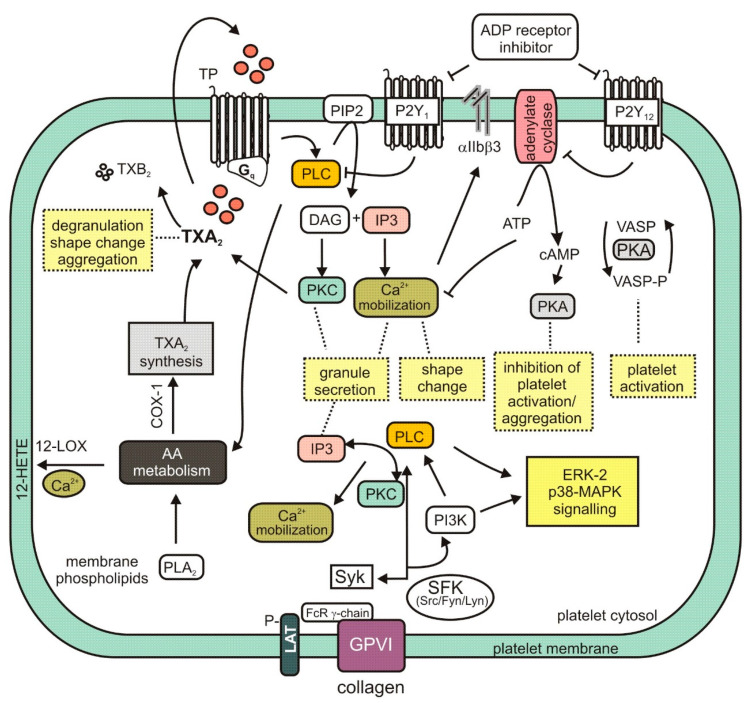
Mechanism of platelet activity modulation showing selected intracellular signaling pathways. 12-HETE, 12-hydroxy-5,8,10,14-eicosatetraenoic acid; 12-LOX, arachidonate 12-lipoxygenase; AA, arachidonic acid; ADP, adenosine 5′-diphosphate; cAMP, adenosine 3′, 5′-cyclic monophosphate; COX-1, cyclooxygenase-1; DAG, 1,2-diacylglycerol; GPVI, glycoprotein VI; IP3, inositol-1,4,5-trisphosphate; LAT, linker for activation of T cells; MAPK, mitogen-activated protein kinase; P2Y_1_, ADP receptor; P2Y_12_, ADP receptor; PI3K, phosphoinositide 3-kinase; PIP2, phosphatidylinositol biphosphate; PKA, protein kinase A; PKC, protein kinase C; PLA2, phospholipase A2; PLC, phospholipase C; SFK, Src-family kinase; Syk, spleen tyrosine kinase; TP, thromboxane receptor; TXA_2_, thromboxane A_2_; TXB_2_, tromboxane B_2_; VASP, vasodilator-stimulated phosphoprotein.

**Table 1 molecules-25-05171-t001:** Steroidal and triterpenoid saponins with antiplatelet and antithrombotic activity (in vitro and in vivo models).

No and Compound Name, Concentration Used	Model	Property	Chemical Structure	Reference
	**Steroidal saponins**			
**(1)** Anemarrhenasaponin A_2_ (50 and 100 µg/mL)	Male Wistar rats (PRP)	Inhibitory effects on ADP-induced platelet aggregation	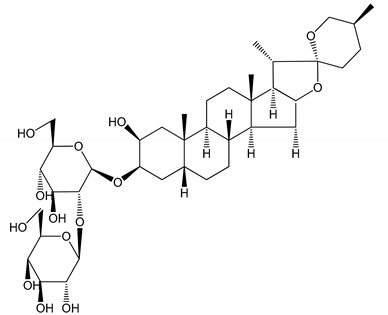	[24]
**(2)** Anemarsaponin B	Male Wistar rats (PRP and washed blood platelets)	Inhibitory effects on ADP-induced platelet aggregation; delayed thromboplastin time	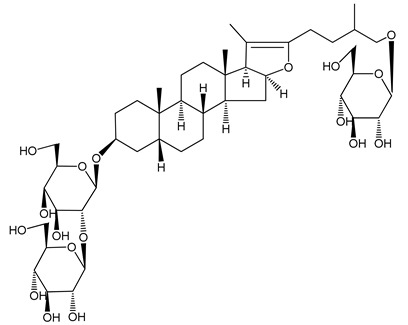	[20]
**(3)** D39 (0.01–1 µM)	Male C57BL/6J mice and HUVEC cells	Inhibition of thrombus formation	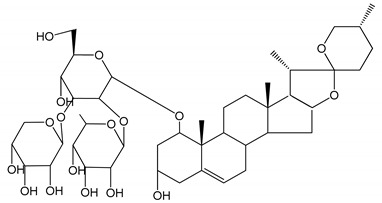	[25]
**(4)** Dioscin (10 mg/kg/day)	Male Kunming mice and Sprague-Dawley rats (PRP)	Antithrombotic effects by improving anticoagulation activity and inhibiting platelet aggregation	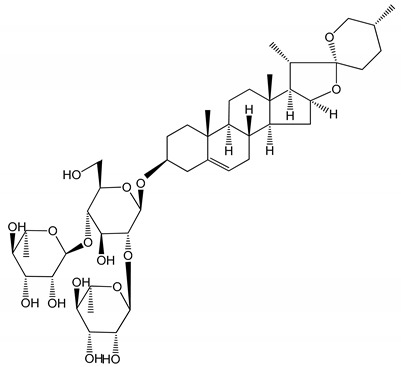	[26]
**(5)** Diosgenyl β-d-galactopyranosyl-(1→4)-β-d-glucopyranoside (25–100 µM)	Male Wistar rats (PRP)	Inhibition of platelet aggregation, antithrombotic activity (prolongation of APTT, inhibition of factor VIII activities)	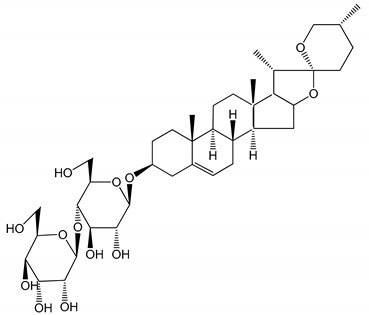	[27]
**(6)** Timosaponin A-III (50, 60 and 100 µg/mL)	Male Wistar rats; male Balb/c mice (PRP and washed blood platelets)	Inhibitory effects on ADP-induced platelet aggregation; delayed thromboplastin time; antithrombotic activities	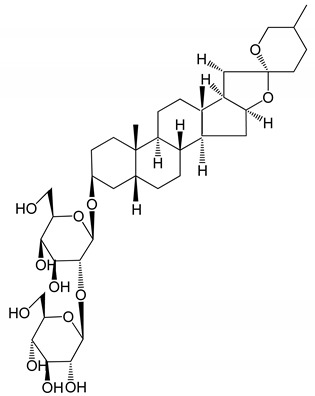	[20,21,24]
**(7)** Timosaponin B-II (50 and 100 µg/mL)	Male Wistar rats; New Zealand white rabbits (PRP)	Inhibitory effects on ADP-induced platelet aggregation; delayed thromboplastin time; antithrombotic activities	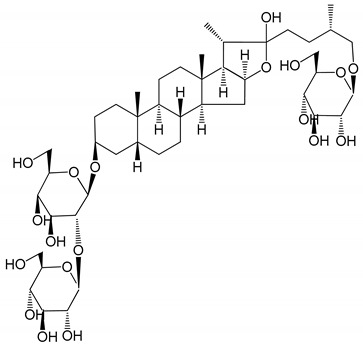	[20,24,28]
	**Triterpenoid saponins**			
**(8)** 2α,3α,19α,23-tetrahydroxyurs-12,20(30)-dien-28-oic acid (1–50 µM)	Rats (PRP)	Inhibitory effects on ADP-induced platelet aggregation	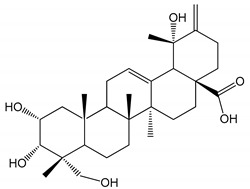	[29]
**(9)** 2α,3α,19α-trihydroxyurs-12-en-28-oic acid 28-O-β-d-xylopyranosyl (1→2)-β-d-glucopyranoside (1–50 µM)	Rats (PRP)	Inhibitory effects on ADP-induced platelet aggregation	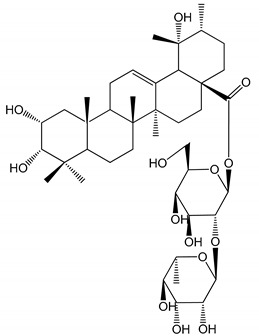	[29]
**(10)** Chikusetsusaponin IVa (0–2000 µM)	Human in vitro studies (washed platelets) male Wistar rats—in vivo studies	Antithrombotic and antiplatelet activity	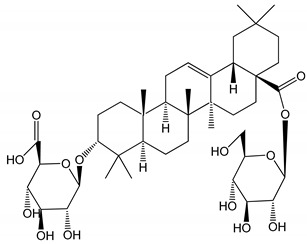	[30]
**(11)** Dihydroginsenoside Rg3 (5–100 µM)	Male Sprague–Dawley rats (washed platelets)	Inhibition of platelet aggregation induced by collagen and thrombin	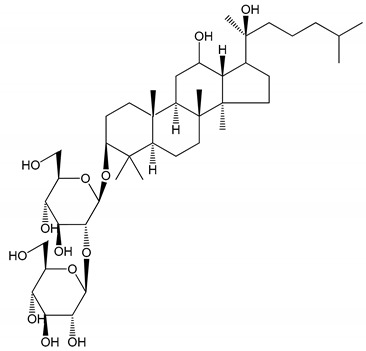	[31]
**(12)** Ginsengoside-2A	Human in vitro studies (PRP and washed platelets)	Decrease of platelet maximum aggregation rate	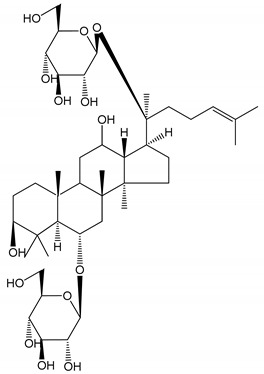	[32]
**(13)** Ginsenoside 20(R)-Rg3 (2.5–10 µM)	Male ICR mice (PRP)	Inhibition of platelet aggregation induced by ADP, collagen, arachidonic acid and U46619 (mimic agent of TXA_2_)	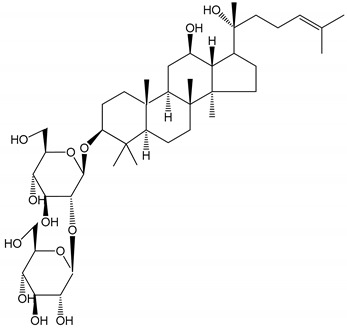	[33]
**(14)** Ginsenoside 20(S)-Rg3 (2.5–10 µM)	Male ICR mice (PRP)	Inhibition of platelet aggregation induced by ADP, collagen, arachidonic acid and U46619 (mimic agent of TXA_2_)	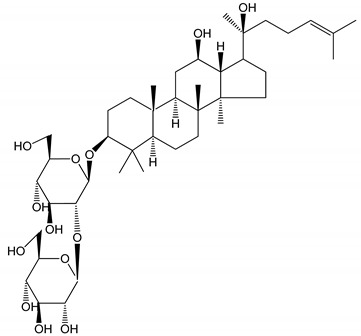	[33]
**(15)** Ginsenoside R1 (10–100 µM)	New Zealand albino rabbits (PRP)	Inhibition of platelet aggregation induced by ADP, collagen, thrombin	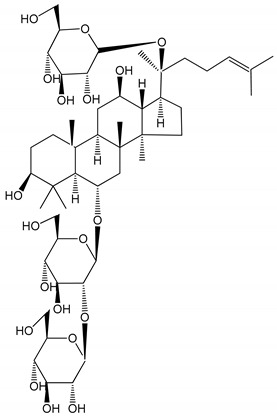	[22]
**(16)** Ginsenoside Ra3 (100 mg/kg/day)	Male Sprague-Dawley rats (PRP)	Antithrombotic effect	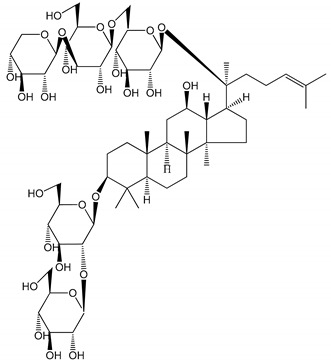	[23]
**(17)** Ginsenoside Rb1 (100 mg/kg/day)	Male Sprague–Dawley rats (PRP)	Antithrombotic effect	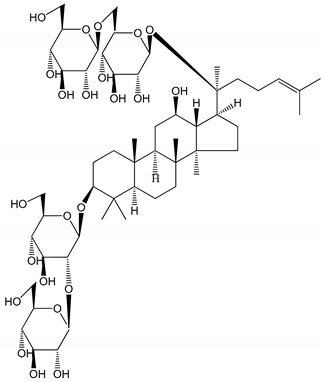	[23]
**(18)** Ginsenoside Rb3	Rabbit	Decrease of platelet maximum aggregation rate	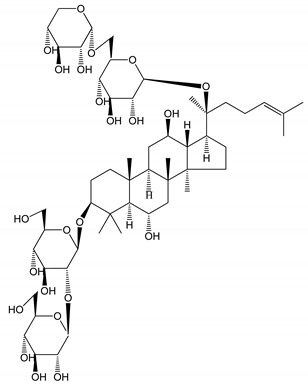	[32]
**(19)** Ginsenoside Rd (100 mg/kg/day)	Male Sprague–Dawley rats (PRP)	Antithrombotic effect	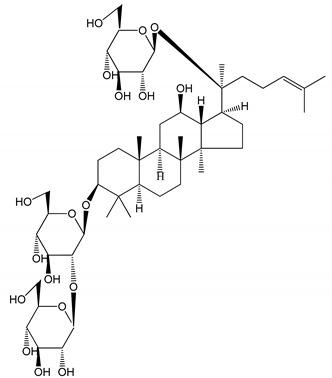	[23]
**(20)** Ginsenoside Rg1 (10–100 µM)	New Zealand albino rabbits (PRP) Human in vitro plasma coagulation assays (PRP)	Inhibition of platelet aggregation induced by ADP, collagen, thrombin Anticoagulation activity	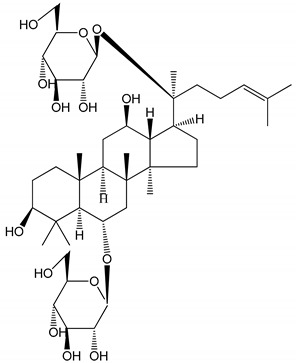	[22] [34]
**(21)** Ginsenoside Rg2 (0.05 mg/mL)	Human in vitro plasma coagulation assays (PRP)	Anticoagulation activity	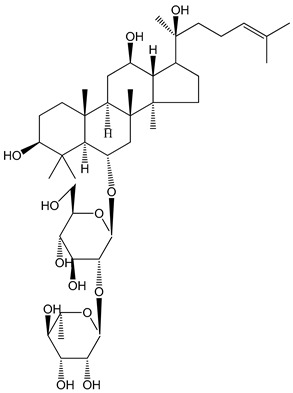	[34]
**(22)** Ginsenoside Rg5 (2.5–10 µM)	Male ICR mice (PRP)	Inhibition of platelet aggregation induced by ADP, collagen, arachidonic acid and U46619 (mimic agent of TXA_2_)	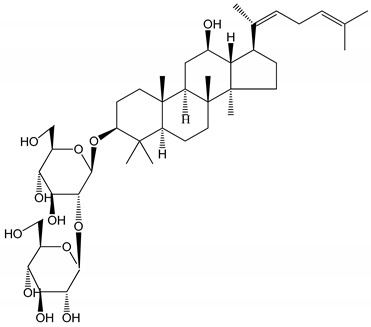	[33]
**(23)** Ginsenoside Re (10–100 µM)	New Zealand albino rabbits (PRP) Male Sprague–Dawley rats (PRP)	Inhibition of platelet aggregation induced by ADP, collagen, thrombin Antithrombotic effect	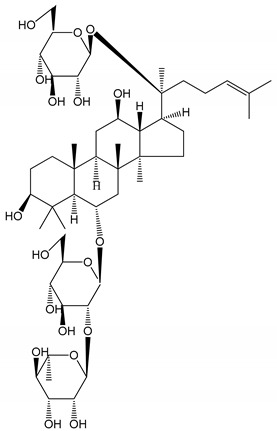	[22] [23]
**(24)** Ginsenosides Rk1 (2.5–10 µM)	Male Sprague–Dawley rats (PRP)	Inhibition of platelet aggregation induced by ADP, collagen, arachidonic acid and U46619 (mimic agent of TXA_2_)	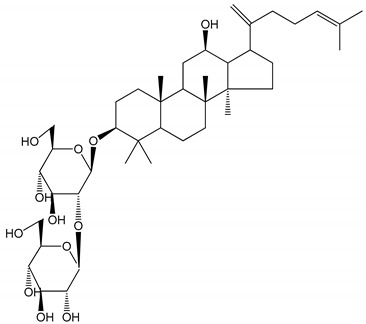	[3,33]
**(25)** Ginsenoside Ro (50–300 µM)	Human in vitro studies (washed platelets)	Inhibition of platelet activation	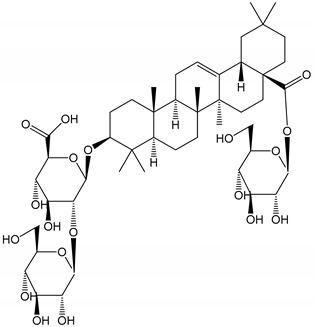	[35]
**(26)** Ginsenoside Rp1 (2.5–100 µM)	Male Sprague–Dawley rats and male C57BL/6J mice (PRP)	Inhibition of platelet activation and thrombus formation	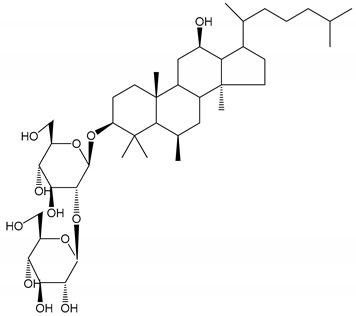	[36]
**(27)** Ginsenoside Rp3 (1.56–50 µM)	Male Sprague–Dawley rats and C57BL/6J mice (washed platelets)	Inhibition of agonist-platelet aggregation and thrombus formation	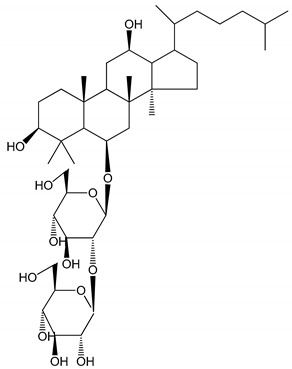	[37]
**(28)** Ginsenoside Rp4 (6.25–50 µM)	Male Sprague–Dawley rats (PRP)	Inhibition of platelet aggregation induced by ADP	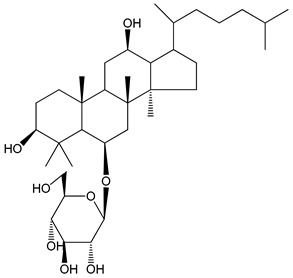	[38]
**(29**–**33)** Glechomanosides A–E (20 µM)	Mice and rabbits (PRP)	Glechomanosides A–E antithrombotic activity Glechomanosides C and D anticoagulant effect	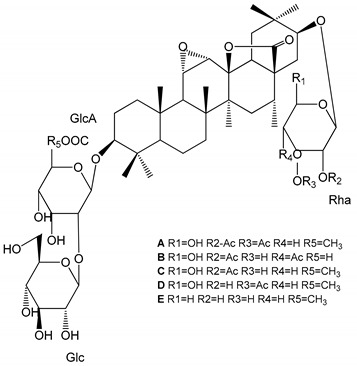	[18]
**(34)** Ilexoside D (10 µM)	Male Sprague–Dawley rats (PRP)	in vivo and in vitro anticoagulant activity	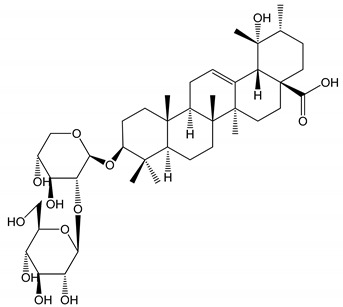	[39]
**(34**–**36)** Ilexosides A, D and J (10 µM)	Male Sprague–Dawley rats (PRP)	Strong inhibitory activities on platelet aggregation induced by thrombin	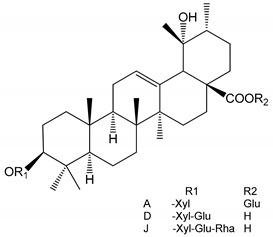	[40]
**(37)** Notoginsenoside Fc (50–800 µM)	Male Sprague–Dawley rats and Kunming mice (washed platelets)	Inhibition of platelet aggregation induced by ADP, collagen, thrombin	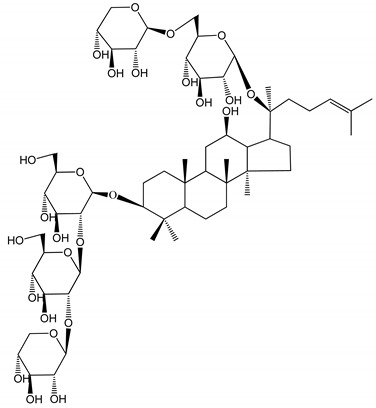	[40]
**(38)** Notoginsenoside Ft1 (50–800 µM)	Male Sprague–Dawley rats and Kunming mice (washed platelets)	Induction of platelet shape change, but not aggregation; haemostatic activity; potentiation of platelet aggregation induced by thrombin	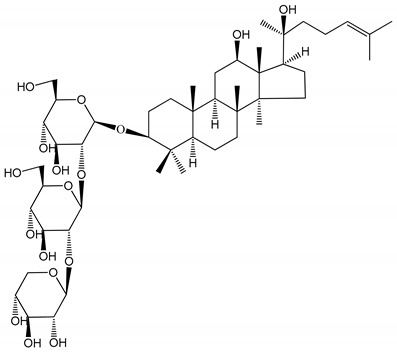	[40]
**(39)** Notoginsenoside R1 (100 mg/kg/day)	Male Sprague–Dawley rats (PRP)	Antithrombotic effect	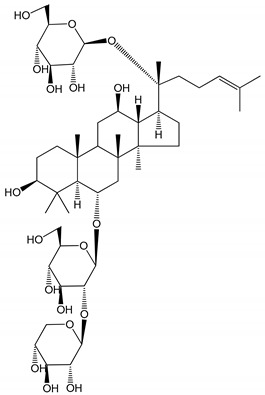	[23]
**(40**–**44)** Sapinmusaponins F-J (1–100 µM)	Human in vitro studies (washed platelets)	Antiplatelet effect, includding anti-aggregatory properties	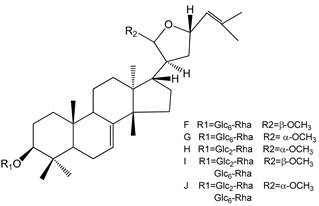	[41]
**(45**,**46)** Sapinmusaponins Q and R (1–50 µM)	Human in vitro studies (washed platelets)	Antiplatelet effect (anti-aggregatory effect) potent than aspirin, IC_50_ ca. 3.4–13.5 mM and 30.5 mM, respectively	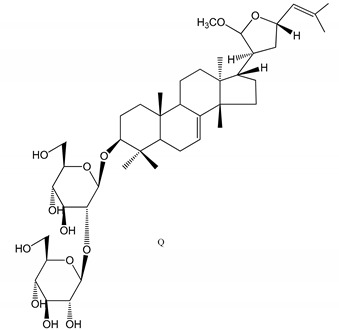 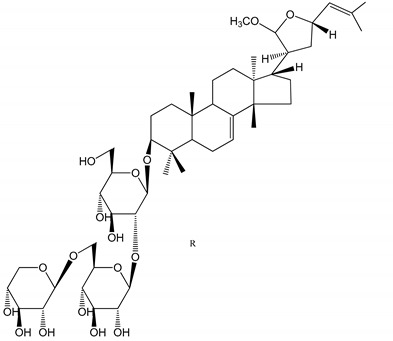	[42]

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
