# Peer review of "Saponins as Modulators of the Blood Coagulation System and Perspectives Regarding Their Use in the Prevention of Venous Thromboembolic Incidents"

_molecules, 2020, doi:10.3390/molecules25215171_

Round 1
Reviewer 1 Report
The article is interesting but does not exhaust the topic.
I recommend that the authors extend the manuscript with additional data that will interest readers.
Please classify saponins according to the type of glycone. Depending on the number of sugar chains included in the molecule, there are: monodesmosides, bidesmosides and tridesmosides, containing 1, 2 or 3 chains, respectively
sugar.
Add information about their various properties in various industries. Describe the prospects for use in various fields. This is sorely lacking in the manuscript. Expand this information.
I believe that the authors should also describe the possibility of using them as precursors in the synthesis sex hormones and corticosteroids.
"GPVI signaling pathway and enzymatic cascades" - this chapter is not well written. It needs to be edited. I believe there is no specificity.
"Notoginsenoside Fc (37) is a novel saponin isolated from Panax notoginseng. The phospholipase 159 C- 2 has been proposed as the focal point of notoginsenoside Fc antiplatelet activity, which is 160 believed to act through the downregulation of DAG, protein kinase C, TXA2 and IP3. In contrast, the 161 notoginsenoside Ft1 is believed to promote platelet aggregation by potentiating signaling by the PLC2-IP3/DAG-(Ca2+ 162 )/PKC-TXA2 pathway (38). Ft1 is isolated from Panax notoginseng in the same 163 way as notoginsenoside Fc [41]." - Nothing follows from this chapter. There is no summary. To change.
"3. Inhibition of tissue factor expression" - And where are the conclusions summarizing this chapter and your own thoughts?
"A study of 41 triterpenoid saponins and sapogenins by Vo et al." - citation should be after the author.
Please insert figures and tables into the text of the manuscript. Reduce or increase accordingly. Remember about the resolution of photos (minimum 300x300 DPI).
Figure 2 has been very poorly described in the text. This should be changed with more details.
Add information on the purification options (biotechnological processes) of saponins and their use.
Author Response
Reviewer comment
The article is interesting but does not exhaust the topic. I recommend that the authors extend the manuscript with additional data that will interest readers.
Please classify saponins according to the type of glycone. Depending on the number of sugar chains included in the molecule, there are: monodesmosides, bidesmosides and tridesmosides, containing 1, 2 or 3 chains, respectively sugar.
Authors response
The “traditional” classification used by us is correct and is widely used in the scientific literature. We have prepared table 1 according to No and compound name, concentration used of saponins with antiplatelet and antithrombotic activity. In our opinion, the new division will destroy all the order.
Reviewer comment
Add information about their various properties in various industries. Describe the prospects for use in various fields. This is sorely lacking in the manuscript. Expand this information.
Authors response
We have not added information on the use of biological properties of saponins in various industries, because in this article we specifically focus on their potential use in the prevention of venous thromboembolic events. In our opinion, introducing new topics will disturb the structure of the manuscript and confuse the reader.
Reviewer comment
I believe that the authors should also describe the possibility of using them as precursors in the synthesis sex hormones and corticosteroids.
Authors response
In this article, we focus only on the specific activity of compounds classified as saponins. Sex hormones and corticosteroids, although they can modulate coagulation system and saponins may be their precursors, are different chemical groups.
Reviewer comment
"GPVI signaling pathway and enzymatic cascades" - this chapter is not well written. It needs to be edited. I believe there is no specificity.
Authors response
In the chapter "GPVI signaling pathway and enzymatic cascades" we indicate the "main players" in the cell signaling pathways through which the activity of selected saponins takes place. This chapter is based on all the literature available today. It deals with aspects of molecular biology and is, in our opinion, written in a manner typical of such issues.
Reviewer comment
"Notoginsenoside Fc (37) is a novel saponin isolated from Panax notoginseng. The phospholipase 159 C-g 2 has been proposed as the focal point of notoginsenoside Fc antiplatelet activity, which is 160 believed to act through the downregulation of DAG, protein kinase C, TXA2 and IP3. In contrast, the 161 notoginsenoside Ft1 is believed to promote platelet aggregation by potentiating signaling by the PLCg2-IP3/DAG-(Ca2+ 162 )/PKC-TXA2 pathway (38). Ft1 is isolated from Panax notoginseng in the same 163 way as notoginsenoside Fc [41]." - Nothing follows from this chapter. There is no summary. To change.
Authors response
In our opinion, the indicated fragment is correctly constructed and fulfills its purpose.
The paragraph deals with the two notoginsenosides Fc and Ft1 and lists the "major players" in the cell signaling pathways through which these saponins can exert antiplatelet activity and modulate platelet aggregation. In the aspect of the molecular basis of the functioning of the coagulation system, this information is important in our opinion. Especially in view of the very small literature on this subject. The reader can also trace in Figure 2 the molecular pathway activated by these notoginsenosides.
Reviewer comment
"3. Inhibition of tissue factor expression" - And where are the conclusions summarizing this chapter and your own thoughts?
Authors response
This chapter is structured like the previous ones. First there is an introduction and later the cell signaling pathway activated by D39 steroidal saponins is discussed. The aim was to identify the "major molecular players" and this was done in the text "It has been shown that the natural D39 (3) steroidal saponin isolated from Liriope muscari (Decne.) LHBailey, down-regulates endothelial TF expression and venous thrombus formation by modulating the PI3K / Akt / GSK3β and NF-κB signaling pathways. It has been proposed that D39 exerts this activity by binding to, and thus deactivating, NMMHC IIA (non-muscular myosin heavy chain IIA), and inhibiting the dissociation of NMMHC II2 from tumor necrosis factor receptor 2 (TNFR2)." We have included a summary and thoughts in the Conclusion section.
Reviewer comment
"A study of 41 triterpenoid saponins and sapogenins by Vo et al." - citation should be after the author.
Authors response
Citation has been corrected.
Reviewer comment
Please insert figures and tables into the text of the manuscript. Reduce or increase accordingly. Remember about the resolution of photos (minimum 300x300 DPI).
Authors response
Figures and table have been inserted. The resolution is correct as required.
Reviewer comment
Figure 2 has been very poorly described in the text. This should be changed with more details.
Authors response
Figure 2 is a visualization of cell signaling pathways activated by selected saponins, detailed described in Sections 3.1. Arachidonic acid pathway and 3.2. GPVI signaling pathway and enzymatic cascades. Figure 2 makes it easier for the reader to trace these processes. All molecular elements mentioned in the text are also in the figure. In our opinion, the text and figure 2 correspond and complement well each other. Additionally, all acronyms have been clarified.
The following text has also been added:
“The elements of the arachidonic acid pathway and GPVI signaling pathway mentioned in the article are presented in Figure 2.”
Reviewer comment
“Add information on the purification options (biotechnological processes) of saponins and their use.”
Authors response
We describe specific saponins, assuming that they are pure chemicals. We focus on biological activity and cell signaling pathways. We believe that biotechnology processes go beyond the topic and can divert attention away from the main issue.
Reviewer 2 Report
The review of Beata Olasa et al. on saponins as antithrombotinc agents recalls many paper on the saponin field. However, it is hard for me to see how saponins are used as antithrombotic. As a reader I would like to know how this compound are used in (chinese) medical practice. Also how an antithrobotic saponin based preparation is standarized.
Several of the listed saponins are claimed as inhibitor of ADP induced aggregation. However in figure 1 no mention of the ADP receptor is given.
The receptor to ADP induced aggregation was discovered in 2001 and it was named P2Y12 receptor, many years after the discovery of thienopyridine Ticlopidine and Clopidogrel. Now many anti-aggregants are based on inhibition of this receptor. (clopidogrel, prasugrel and ticagrelor)
1: Hollopeter G, Jantzen HM, Vincent D, Li G, England L, Ramakrishnan V, Yang RB, Nurden P, Nurden A, Julius D, Conley PB. Identification of the platelet ADP receptor targeted by antithrombotic drugs. Nature. 2001 Jan 11;409(6817):202-7. doi: 10.1038/35051599. PMID: 11196645.
2: Foster CJ, Prosser DM, Agans JM, Zhai Y, Smith MD, Lachowicz JE, Zhang FL, Gustafson E, Monsma FJ Jr, Wiekowski MT, Abbondanzo SJ, Cook DN, Bayne ML, Lira SA, Chintala MS. Molecular identification and characterization of the platelet ADP receptor targeted by thienopyridine antithrombotic drugs. J Clin Invest. 2001 Jun;107(12):1591-8. doi: 10.1172/JCI12242. PMID: 11413167; PMCID:PMC200194.
This is a major drawback in a review written in 2020. I think that a critical paragraph on this receptor should be written and that figure 1 should be modified to give the state of the art.
Moreoverin the coagulation part, there is no mention of the coagulation cascade (prothrombin, coagulation factors) and of anti-vitamin K (Glutamyl-carboxylase and VKOR).
The description and description of content of the many saponin papers seems correct.
Thus I think that the paper must be very much edited in function of the above remarks. A review must give the state of the art, and a pharmacological review must allow the reader to understand the usage of claimed drugs.
Author Response
Reviewer comment
The review of Beata Olasa et al. on saponins as antithrombotinc agents recalls many paper on the saponin field. However, it is hard for me to see how saponins are used as antithrombotic. As a reader I would like to know how this compound are used in (chinese) medical practice. Also how an antithrobotic saponin based preparation is standarized.
Authors response
The article only shows the anti-platelet and antithrombotic activity of selected saponins (Table 1). This activity relates to the cell signaling pathways and the aspect of molecular biology. This, in turn, may lead to the development of effective anticoagulants. Certain anticoagulant plants are likely used in Chinese medicine, but we are writing about pure chemicals. Plants, on the other hand, contain many different compounds with different activities. We propose saponins as therapeutic tools for the future.
Reviewer comment
Several of the listed saponins are claimed as inhibitor of ADP induced aggregation. However in figure 1 no mention of the ADP receptor is given.
Authors response
We think that this remark applies to figure 2. Figure 1 shows only the structures and distribution of the saponins.
We have added the P2Y12 receptor and ADP receptor inhibitor to the figure 2. We also associated P2Y12 with inhibition of adenylate cyclase activity.
Reviewer comment
The receptor to ADP induced aggregation was discovered in 2001 and it was named P2Y12 receptor, many years after the discovery of thienopyridine Ticlopidine and Clopidogrel. Now many anti-aggregants are based on inhibition of this receptor. (clopidogrel, prasugrel and ticagrelor)
1: Hollopeter G, Jantzen HM, Vincent D, Li G, England L, Ramakrishnan V, Yang RB, Nurden P, Nurden A, Julius D, Conley PB. Identification of the platelet ADP receptor targeted by antithrombotic drugs. Nature. 2001 Jan 11;409(6817):202-7. doi: 10.1038/35051599. PMID: 11196645.
2: Foster CJ, Prosser DM, Agans JM, Zhai Y, Smith MD, Lachowicz JE, Zhang FL, Gustafson E, Monsma FJ Jr, Wiekowski MT, Abbondanzo SJ, Cook DN, Bayne ML, Lira SA, Chintala MS. Molecular identification and characterization of the platelet ADP receptor targeted by thienopyridine antithrombotic drugs. J Clin Invest. 2001 Jun;107(12):1591-8. doi: 10.1172/JCI12242. PMID: 11413167; PMCID:PMC200194.
This is a major drawback in a review written in 2020. I think that a critical paragraph on this receptor should be written and that figure 1 should be modified to give the state of the art.
Authors response
The biological activity of some saponins is accomplished by P2Y12, and this is the only thing that we described, mainly in Table 1.
We do not describe this mechanism as something new, and we do not cite old literature data. This signaling pathways is marked in figure 2 for the convenience of the reader, and we have limited ourselves to that only.
Reviewer comment
Moreoverin the coagulation part, there is no mention of the coagulation cascade (prothrombin, coagulation factors) and of anti-vitamin K (Glutamyl-carboxylase and VKOR).
Authors response
We have added short information about it;
“Some studies report the pharmacological properties of single and specific saponins with antithrombotic, anti-platelet aggregation and anticoagulation effects [20-23]. Table 1 summarizes the results of recent in vitro and in vivo studies, based on various models, on the antiplatelet and antithrombotic activity of individual saponins. For example, some saponins have effect on the coagulation cascade, including prothrombin and other coagulation factors [18,26,27]”
Reviewer comment
The description and description of content of the many saponin papers seems correct.
Reviewer comment
Thus I think that the paper must be very much edited in function of the above remarks. A review must give the state of the art, and a pharmacological review must allow the reader to understand the usage of claimed drugs.
Authors response
There are no saponin-based anticoagulants available yet. We present only saponins with such activity and the molecular basis of their action.
Round 2
Reviewer 2 Report
The authors have written a review on saponin use as modulators of blood coagulation.
It is a good review on saponins. I stiil think that the coagulation mechanisms should have been more developped, mentioning the Vitamin K dependent gamma-glutamyl carboxylase, and the ADP stimulated pathway. The authors simply added P2Y12 receptor in figure 2 and once in the text without giving any reference. Thus I think this part is truncated.
I like to mention that these two mechanisms are the base of the most sold anticoaggulants before 2012 (VKA and antiplatelets.
However the authors state that saponins act on ADP secretion. OK!
Thus on the point of vue of saponins only, the paper could be accepted.
Author Response
Reviewer comment
It is a good review on saponins. I stiil think that the coagulation mechanisms should have been more developped, mentioning the Vitamin K dependent gamma-glutamyl carboxylase, and the ADP stimulated pathway. The authors simply added P2Y12 receptor in figure 2 and once in the text without giving any reference. Thus I think this part is truncated.
I like to mention that these two mechanisms are the base of the most sold anticoaggulants before 2012 (VKA and antiplatelets.
However the authors state that saponins act on ADP secretion. OK!
Thus on the point of vue of saponins only, the paper could be accepted.
Authors response
According to the Reviewer's comments, we expanded the issues related to ADP receptor inhibition:
- we changed Figure 2 again. We also added the P2Y1 receptor as a target for ADP receptor inhibitors.
- at the end of chapter 3.2 “GPVI signaling pathway and enzymatic cascades” we have added a text on the molecular basis of the signaling pathways in which ADP receptors are involved (lines 173-180):
“Several of the saponins listed in Table 1 have the ability to inhibit ADP-induced platelet aggregation. These compounds are therefore antagonists of ADP-activated transmembrane receptors such as P2Y1 and P2Y12. Both of these receptors interact with G proteins, but their downstream effectors are in different signaling pathways. The activation of a P2Y1 ADP receptor, which belongs to the Gq protein-coupled receptors, leads to the activation of phospholipase C (PLC), whereas activation of the P2Y12 receptor coupled to the Gi protein triggers adenylate cyclase activity. It has been suggested that activation of both receptors is required for a complete platelet response to ADP [54,55]. Figure 2 shows the potential effectors inhibited by ADP receptor inhibitors.”
and in Conclusion section (lines 288-291)
“Platelets are activated in many ways, so even with efficient inhibition of platelets by ADP inhibitors such as clopidogrel, prasugrel, or ticagrelor, their activation may occur through other molecular pathways [84]. Whereas, saponins as a group of chemical compounds offer broad anticoagulant activity through various cell signaling pathways.”
- we have added three references:
Thibeault, P. E., Ramachrandran, R. Biased signalling in platelet G-protein‐coupled receptors. Can J. Physiol. Pharmacol. 2020.
Offermanns, S. Activation of platelet function through G protein–coupled receptors. Circ. Res. 2006, 12, 1293-1304.
Laine, M., Paganelli, F., Bonello, L. P2Y12-ADP receptor antagonists: days of future and past. World J. Cardiol. 2016, 8.
Regarding the clotting process and the share of vitamin K and gamma-glutamyl carboxylase, we added the text in Conclusion section (lines 296-302):
“The ability of saponins to increase membrane permeabilization is also an important issue. On the one hand, it can increase the bioavailability of drugs or vitamins, and on the other hand, it may lead to pro-thrombotic activity. This is especially important in the prothrombin activation process in which vitamin K serves as an essential effector for γ-glutamyl carboxylase, an enzyme that catalyzes the carboxylation of glutamic acid residues in prothrombin. Saponins are substances which increase the permeabilization of the microsomal membrane in this process and thus facilitate the coagulation [85-87].”
- we have added three references:
Ayombil, F., Camire, R. M. Insights into vitamin K-dependent carboxylation: home field advantage. Haematologica 2020, 105, 1996-1998.
Sun, F., Ye, C., Thanki, K., Leng, D., van Hasselt, P. M., Hennink, W. E., van Nostrum, C. F. Mixed micellar system stabilized with saponins for oral delivery of vitamin K. Colloids Surf. B: Biointerfaces 2018, 170, 521-528.
Tollefsen, S., Wierød, L., Skotte, A., Rob, J. A., Helgeland, L. Saponin permeabilization of rough microsomes from rat liver reveals a novel prothrombin pool. Biochim. Biophys. Acta 2001, 1526, 249-256.
We found no literature data that would allow to combine saponins, vitamin K and gamma-glutamyl carboxylase directly. Since we did not describe the mechanisms related to platelet aggregation / activation and clotting as such in the article, we could not write about this mechanism either. In our opinion, it would be an inconsistency.
However, we found interesting information about the permeabilization properties of saponins and added it.